# Rice (*Oryza sativa* L.) Grain Size, Shape, and Weight-Related QTLs Identified Using GWAS with Multiple GAPIT Models and High-Density SNP Chip DNA Markers

**DOI:** 10.3390/plants12234044

**Published:** 2023-11-30

**Authors:** Nkulu Rolly Kabange, Gamenyah Daniel Dzorkpe, Dong-Soo Park, Youngho Kwon, Sais-Beul Lee, So-Myeong Lee, Ju-Won Kang, Seong-Gyu Jang, Ki-Won Oh, Jong-Hee Lee

**Affiliations:** 1Department of Southern Area Crop Science, National Institute of Crop Science, Rural Development Administration (RDA), Miryang 50424, Republic of Korea; parkds9709@korea.kr (D.-S.P.); kwon6344@korea.kr (Y.K.); pappler@korea.kr (S.-B.L.); olivetti90@korea.kr (S.-M.L.); kangjw81@korea.kr (J.-W.K.); sgjang0136@korea.kr (S.-G.J.); ohkw1004@korea.kr (K.-W.O.); 2Council for Scientific and Industrial Research (CSIR), Crops Research Institute, Kumasi 3785, Ghana; gamenyahdaniel@gmail.com

**Keywords:** SNP chip DNA marker, multiple GAPIT models, GWAS, genomic selection, grain traits, rice

## Abstract

This study investigated novel quantitative traits loci (QTLs) associated with the control of grain shape and size as well as grain weight in rice. We employed a joint-strategy multiple GAPIT (Genome Association and Prediction Integrated Tool) models [(Bayesian-information and Linkage-disequilibrium Iteratively Nested Keyway (BLINK)), Fixed and random model Circulating Probability Uniform (FarmCPU), Settlement of MLM Under Progressive Exclusive Relationship (SUPER), and General Linear Model (GLM)]–High-Density SNP Chip DNA Markers (60,461) to conduct a Genome-Wide Association Study (GWAS). GWAS was performed using genotype and grain-related phenotypes of 143 recombinant inbred lines (RILs). Data show that parental lines (Ilpum and Tung Tin Wan Hein 1, TTWH1, *Oryza sativa* L., ssp. *japonica* and *indica*, respectively) exhibited divergent phenotypes for all analyzed grain traits), which was reflected in their derived population. GWAS results revealed the association between seven SNP Chip makers and QTLs for grain length, co-detected by all GAPIT models on chromosomes (Chr) 1–3, 5, 7, and 11, were *qGL1-1^BFSG^* (AX-95918134, Chr1: 3,820,526 bp) explains 65.2–72.5% of the phenotypic variance explained (PVE). In addition, *qGW1-1^BFSG^* (AX-273945773, Chr1: 5,623,288 bp) for grain width explains 15.5–18.9% of PVE. Furthermore, BLINK or FarmCPU identified three QTLs for grain thickness independently, and explain 74.9% (*qGT1^Blink^*, AX-279261704, Chr1: 18,023,142 bp) and 54.9% (*qGT2-1^Farm^*, AX-154787777, Chr2: 2,118,477 bp) of the observed PVE. For the grain length-to-width ratio (LWR), the *qLWR2^BFSG^* (AX-274833045, Chr2: 10,000,097 bp) explains nearly 15.2–32% of the observed PVE. Likewise, the major QTL for thousand-grain weight (TGW) was detected on Chr6 (*qTGW6^BFSG^*, AX-115737727, 28,484,619 bp) and explains 32.8–54% of PVE. The *qTGW6^BFSG^* QTL coincides with *qGW6-1^Blink^* for grain width and explained 32.8–54% of PVE. Putative candidate genes pooled from major QTLs for each grain trait have interesting annotated functions that require functional studies to elucidate their function in the control of grain size, shape, or weight in rice. Genome selection analysis proposed makers useful for downstream marker-assisted selection based on genetic merit of RILs.

## 1. Introduction

Rice (*Oryza sativa* L.) remains a staple food crop for more than half of the world’s population [1,2], and serves as an important source of calories for human health and fitness [3]. Rice consumption is increasing faster than any other cereals [4]. Considering the gradual increase in global population, predicted to reach about 9.8 billion by 2050 [2,5], the production of rice must increase to over 852 million tons by 2035 [6] to meet the growing food demands, in addition to environmental challenges amplified by climate change effects that exacerbate food insecurity. In many countries, rice is consumed as a whole grain and can be processed into different types of food. Rice grain size, shape, appearance, and quality directly influence the market value [7,8]. Based on grain size, preferences for its quality vary from one region to another across the globe. Rice grain size and shape determine the milling efficiency and grain recovery, which in turn influence its price.

Many rice-breeding programs have long been oriented toward the development of high-yielding and disease-resistant rice varieties [9,10,11]. As part of the diversification process to address the rising food demands in terms of quantity and quality, the trend in rice breeding has shown a keen interest in the quality of grains coupled with productivity [2,12,13,14].

The phenotype of rice appearance is determined by grain shape (length, width, and thickness), translucency, and thousand-grain weight [15,16]. At the genetic level, studies have identified genes associated with the control of grain size, shape, and weight of rice, which result from a complex interaction between major and minor quantitative trait loci (QTLs) [17]. For instance, grain size has a high heritability, and is controlled by a wide range of major [15] and minor QTLs [18], which are influenced by several factors, including the environment and genetic variation. Several grain traits-related genetic factors have been investigated in natural populations, intervarietal lines, mutant populations [6], doubled haploid (DH) lines [19], near-isogenic lines (NILs) [20] and recombinant inbred lines (RILs) [21]. Many genetic studies employed molecular markers, such as amplified fragment length polymorphism (AFLP), simple sequence repeat (SSR), or microsatellite (RM) [6,22], single nucleotide polymorphism (SNP) [23], and restriction fragment length polymorphism (RFLP) [24], to improve grain appearance and other agronomically important grain traits [2].

Likewise, genome-wide association study (GWAS), linkage mapping, and omics tools were used to investigate genetic loci controlling the complex grain-quality traits such as grain shape (appearance), milling quality, nutritional quality, and eating and cooking qualities [25]. Furthermore, several grain traits-related QTLs have been reported, and mapped to all chromosomes of rice. Among them, we could mention *grain size 3* (*GS3*), controlling both grain length and weight [26,27]; *grain width 2* (*GW2*) [28]; *wide and thick grain* (*OsOTUB1*/*WTG1*) [29]; *GS5*, which regulates a putative serine carboxypeptidase (SCP) that specifically affects grain width and filling [30]; *GS2*, which encodes a growth-regulating factor 4 (GRF4) that governs grain length and width [31]; *grain length 3* (*GL3.1*/*qGL3*), which acts on a putative protein phosphatase and influences grain length, width and weight [32,33]; and *GW5*/*qSW5*, a calmodulin-binding protein responsible for grain width and weight [34,35]. Another set of genetic loci such as *thousand-grain weight 3* (*TGW3*) and *TGW6* (encodes the auxin signaling pathway that regulates grain weight) [36,37], or *GW6* (encodes a gibberellin-regulated GAST family protein that controls grain width and weight) [38], are proposed to be involved in the regulation of grain-related traits in rice. Likewise, *GLW7* and *GW8* not only encode the *OsSPL13* and *OsSPL16* transcription factors but also contribute to grain size in rice [39,40,41]. All these QTLs function as either positive or negative regulators in various signaling pathways, including the G-protein signaling, ubiquitin-proteasome, phytohormone, and transcriptional regulation pathways that influence cell division and endosperm development, among other processes and characteristics [42,43,44].

This study aimed at investigating novel QTLs controlling the grain size, shape, and weight of rice using an RIL population consisting of 143 lines derived from a cross between *indica* and *japonica* cultivars. To achieve that, a joint strategy employing GWAS with multiple Genome Association and Prediction Tool (GAPIT) models and high-density SNP Chip DNA markers was used.

## 2. Results

### 2.1. Diverging Grain Phenotypes between Parental Lines and RILs

Considering that they belong to the most cultivated subspecies of rice (*Oryza sativa* L. ssp. *indica* and *japonica*), parental lines (Ilpum and Tung Tin Wan Hein 1, TTWH1) exhibited differential phenotypes for all analyzed rice grain size, shape, and weight-related traits, as expected (Figure 1A–D). The *japonica* parent (Ilpum) showed relatively shorter grains (Figure 1A) and lower LWR (Figure 1B) while having larger grains (grain width, GW) (Figure 1A) and higher thousand-grain weight (TGW, Figure 1C). In contrast, the *indica* TTHW1 had longer grains (Figure 1A) and higher LWR (Figure 1B), but thinner grains and lower TGW (Figure 1C). However, although we recorded an arithmetic or numerical difference between the grain thickness of Ilpum (thinner grains) compared to that of TTWH1 (thicker grains), a non-significant statistical difference was observed.

In addition, we observed a positive (right) skewness for GL (Figure 2A, *p* < 0.001) and LWR (Figure 2G, *p* < 0.001), as supported by the Shapiro–Wilk and d’Agostino and Pearson normality tests (Appendix A). However, GW (Figure 2D, *p* < 0.001), GT (Figure 2J, *p* < 0.001), and TGW (Figure 2M, *p* < 0.01) exhibited a negative (left) skewness. As displayed in Figure 2B,E,H,K,N, a total shift (Ilpum-like pattern) in GL, GW, LWR, GT, and TGW of the RIL population was observed. Meanwhile, nearly 82.5% and 97.9% of the RIL population exhibited relatively short grains (Ilpum-like) and LWR value, respectively, against 17.5% and 2.1% having long grains and LWR (TTWH1-like phenotype). Likewise, the assessment of symmetric grain traits phenotypic distribution using the quantile–quantile (Q–Q) plot supports the observed negative or positive skewness of analyzed traits (Figure 2C,F,I,L,O).

### 2.2. Relatedness, Correlation, Heritability, and Genomic Selection

We constructed a kinship matrix to assess the relatedness of the RIL population. As indicated in Figure 3A, based on the color pattern in the heat map, the majority of genotypes of RILs used in this study are diverse and not closely related. The density of SNP Chip DNA markers across all 12 chromosomes of rice is provided in Figure 3B. Figure 3C shows that RILs were grouped into three clusters based on their recorded grain size, shape, and weight phenotypes. Principal Component Analysis (PCA) results suggest that PC1 (55.6%), PC2 (30.8%), and PC3 (9.8%) explain 96.2% of the proportion of phenotypic variance of the RILs population.

Furthermore, to understand the proportion of variance explained by the individuals’ breeding values for the target traits, we estimated the narrow sense heritability (h^2^) of traits. Data in Figure 3D indicate that GL had an h^2^ of 0.915, while GW (Figure 3E), GT (Figure 3F), grain LWR (Figure 3G), and TGW (Figure 3H) showed an h^2^ of 0.885, 0.852, 0.454, and 0.831, respectively.

To further gain insights and assess the genetic merits of individuals in the RIL population for the target traits, we performed a genomic selection analysis based on the MLM (gBLUP) method, which is known to have a high prediction accuracy for genomic estimated breeding value (GEBV) for traits controlled by a large number of genes. The resulting output of the genomic selection analysis shows the predicted and observed GEBV of individuals in the RIL population for TGW in the reference (Figure 3I) and inference (Figure 3J) groups.

Correlation analysis is useful for understanding the relationship between two variables or identifying possible inputs for testing changes in a dependent variable while holding other variables constant. To explore the relationship between traits, we conducted a correlation analysis using the Pearson correlation method. Results in panels A and D in Figure 4 reveal a weak positive correlation between GL (R^2^ = 0.153 ***) or GT (R^2^ = 0.205 ***) and TGW. In contrast, panel B in Figure 4 suggests the existence of a strong positive correlation between GW and TGW (R^2^ = 0.392 ***).

### 2.3. Identified QTLs for Grain Size and Shape in Rice through GWAS with Multiple GAPIT Models

We conducted a GWAS employing multiple GAPIT models, with enhanced power and accuracy for genome association, to investigate novel genetic loci for grain size and shape, and TGW, which are essential components of rice yield. GWAS results identified 43 QTLs (all grain traits considered and GAPIT models cumulated), of which 10 QTLs were associated with GL, 14 with GW, 3 with GT, 8 with LWR, and 8 with TGW (Table 1 and Appendix A). From these results, we were interested to see co-detected QTLs with the highest contribution to the trait values. In this regard, we found that among the detected GL-related QTLs, seven out of ten were co-detected by both BLINK and FarmCPU GAPIT models, and mapped on Chr 1–3, 5, 7, and 11 (Figure 5A–E). Among them, four QTLs were co-detected by all GAPIT models, and two QTLs by three GAPIT models, of which AX-95918134 (*qGL1-1^BFSG^*, Chr1, 3,820,526 bp, allelic effect: TTWH1) explains 65.5–72.5% of the observed phenotypic variance (PVE; Figure 5A, Table 1 and Appendix A).

Likewise, among the fourteen QTLs associated with grain width (GW) identified here, six QTLs were co-detected by 2–4 GAPIT models, where *qGW1-1^BFSG^* (AX-273945773, Chr1:5,623,288 bp) explains a PVE of about 15.5–18.9%, and the allele from Ilpum contributed to the trait value. In addition, the GW-related QTL *qGW6-1^Blink^* coincides with the *qTGW6^BFG^* locus for TGW, which can be noted as *qGW6-1^Blink^/TGW6^BFSG^* (AX-115737727, Chr6: 28,484,619 bp) (Figure 5B, Table 1). Other QTLs for GW are located on Chr1–3, 6, 8, and 12. The *qGW6-2^FSG^*, *qGW8^FSG^*, and *qGW12^FSG^* were co-detected by FarmCPU, SUPER, and GLM.

Concerning GT, three QTLs (AX-279261704, Chr1: 18,023,142 bp, PVE 74.9%; AX-154787777, Chr2: 2,118,477 bp, PVE 54.9%, and AX-154913392, Chr2: 25,105,471 bp, PVE 5.3%) were detected by BLINK (*qGT1^Blink^*) and FarmCPU (*qGT2-1^Farm^* and *qGT2-2^Farm^*) (Figure 5C, Table 1). Meanwhile, four out of eight QTLs associated with LWR, were co-detected by 2–3 GAPIT models, with *qLWR2^BFSG^* (AX-274833045, Chr2: 10,000,097 bp, allelic effect: TTHW1) being the only one co-detected by all four GAPIT models. However, *qLWR6-1^FSG^* (AX-115851421, Chr6: 10,178,858 bp, recorded the highest PVE value (PVE 30.5%) (Figure 5D, Table 1).

Thousand-grain weight (TGW) is an important component of rice yield, and is determined by several factors, including GL, GW, and GT, among others. Our data in Table 1 shows that two out of eight QTLs associated with the control of TGW were co-detected by all four GAPIT models; meanwhile, BLINK and FarmCPU co-detected two others. The SNP Chip marker AX-115737727, linked to the *qTGW6^BFSG^* QTL (Chr6: 28,484,619 bp), which coincides with GW QTL *qGW6-1^Blink^* as indicated earlier, is here regarded as the major QTL for TGW identified by this study, considering its co-detection by all GAPIT models used and its high PVE value. The latter is followed by *qTGW2-1^BF^* (AX-279699609, Chr2 (10,805,604 bp, PVE 18.7–27.9%) and *qTGW3-2^Farm^* (AX-123153600, Chr3: 7,887,961 bp, PVE 13.9% (Figure 5E, Table 1 and Appendix A). Furthermore, Figure 6A,D,G display the plots of LOD values, while Figure 6B,E,H show the estimated effect of detected QTLs associated with the control of GL, GW, and TGW in rice. Likewise, the contribution of the identified QTLs to the trait values is displayed in Figure 6C,F,I.

### 2.4. Putative Candidate Genes Harbored by Grain Traits-Related QTLs

Following the detection of major QTLs associated with the control of target grain size, shape, or weight-related traits, we were interested in unraveling the identity of genes harbored by these QTLs. To achieve that, we used the known physical positions of associated SNP Chip DNA markers co-detected by both BLINK and FarmCPU, in the rice genome database (http://rice.uga.edu/cgi-bin/gbrowse/rice/#search, accessed on 1 September 2023). From data in Table 2, we can see that genes harbored by *qGL1-1^BFSG^* are proposed to be involved in post-embryonic development, reproduction, and/or signal transduction, secondary metabolic (*Os01g07880* and *Os01g07930*, encoding a HY5 and Zinc finger transcription factors, respectively). In the same region, genes associated with transport events (*Os01g07870*, encoding an ATP binding cassette (ABC) transporter), protein modification process (*Os01g07920*) or cellular homeostasis (*Os01g07950*, encoding a glutaredoxin subunit II), protein-binding activities (*Os01g07980*, encoding an Ankyrin repeat domain), or response to abiotic stimuli (*Os01g07910*, encoding NADH-cytochrome b5 reductase) are found.

The *qGW1-1^BFSG^* region (associated with the control of grain width in rice) harbors genes with similar annotated functions to those found in *qGL1-1^BFSG^*. The latter includes the *Os01g10580* (encoding a B-box (BBX) zinc finger transcription factor protein) proposed to be involved in post-embryonic development, cellular component organization, or secondary metabolic process; the *Os01g10590* (*OsFTL8*, encoding an FT-like 8 homologous to flowering locus T gene) involved in flower development and reproduction; the *Os01g10550* (*OsDEFL35*, encoding Defensin-like DEFL protein), *Os01g10600* (Aquaporin), or *Os01g10610* (encoding a Brassinosteroids-regulated transcription factor BES1/BZR1 protein) involved in protein binding and transport activity, respectively.

Likewise, in the *qGT1-1^Blink^* (for grain thickness on Chr1), the *Os01g32930* gene (encoding an SGS domain-containing protein) is proposed to be involved in embryo development, reproduction, or post-embryonic development, among others. The *qGT1-1^Blink^* also harbors genes encoding transcription factors (*Os01g32920*, ZOS1-08, a C2H2 zinc finger TF), transport-related proteins (*Os01g32880*, AP-3 complex protein DnaJ), or protein metabolic process (*Os01g32800*, a proteasome subunit, PINT motif (Proteasome, Int-6, Nip-1 and TRIP15)). In the same way, *qGT2-1^Farm^* carries genes associated with lipid metabolic process, multicellular organization, or flower development (*Os02g04690*, *Os02g04725*, *OsSPL3* TF (*Os02g04680*), or transcriptional regulatory event (*Os02g04640*, myeloblastosis (MYB)-like DNA binding domain), etc.

Like in the case of other genetic loci, putative candidate genes were pooled from QTLs co-detected by at least two GAPIT models. Otherwise, independent QTLs, detected by either BLINK, FarmCPU, or GLM, with the highest PVE value were considered. Thus, in the case of LWR, *qLWR2-1^BFSG^* (AX-274833045, Chr2: 10,000,097 bp (PVE 15.2% (BLINK) or 32.9% (FarmCPU)) was retained to uncover the identity of putative candidate genes. In this region (*qLWR2-1^BFSG^*), a set of genes encoding interesting annotated predicted functions are found. We could mention the VHS (VPS-27, Hrs, and STAM) and GAT (GGA and Tom1, *Os02g17350* responsible for ubiquitin binding and ubiquitination); the *Os02g17380*, encoding pentatricopeptide repeat (PPR) domain-containing protein associated with the restoration of fertility (Cytoplasmic male sterility, CMS); the restoration of fertility 2 (*Rf2, Os02g17380*, encoding a mitochondrial glycine-rich protein) in LD-CMS; the *Os02g17390* (encoding 3-hydroxyacyl-CoA dehydrogenase [45]), involved in flower development or multicellular organismal development; and the Tesmin/TSO1-like transcription factor (*Os02g17460*).

Concerning TGW, the major QTL (AX-115737727 *qGW6-1^Blink^/qTGW6^BFG^*), co-detected by all GAPIT models, harbors genes such as *Os06g46910* coding for a ZOS6-07 C2H2 zinc finger transcription factor, *Os06g46920* (encoding dihydroflavonol-4-reductase, associated with fatty acid catabolism, gibberellin biosynthesis and signaling, or seed dormancy). In the same QTL region (*qTGW6^BFG^*), the *Os6bglu25* gene (*Os06g4930*, encoding a beta-glucosidase homologue proposed to be involved in carbohydrate metabolism), *Os06g46950* (encoding an EF hand protein) associated with anatomical structure morphogenesis, cell differentiation or cellular component organization, are found.

Considering the genetic variability between the *japonica* and *indica* rice subspecies, we were interested to see the degree of similarity of genes found in major QTLs for grain traits identified in Table 2. To achieve that, the coding sequence (CDS) of each gene in the *japonica* group were aligned with their orthologues in the *indica* group. Results in Table 2 (Column 7, CDS *japonica* vs. *indica*) reveal mutation sites (deletion or substitution) in a set of genes, while others showed a 100% similarity between the two subspecies.

**Table 2 plants-12-04044-t002:** Candidate genes harbored by *qGL1-1^BFSG^*, *qGW1-1^BFSG^*, *qGT1^Blink^* and *qGT2-1^Farm^*, *qLWR2-1^BFSG^*, and *qTGW6^BFSG^* loci.

No.	*Japonica*/*Indica*	Description	Biological Process	Molecular Function	Cellular Component	CDS *Japonica* vs. *Indica*/Similar Report
	*qGL1-1^BFSG^*	Chr1:3804000..3883000				
1	*Os01g07870*	ATP-bindinc cassette (ABC) transporter family protein, Peroxidase 56	Transport	Hydrolase activity, transporter activity	Extracellular region, integral component of membrane, vacuole	-; [46]
2	*Os01g07880*	*OsbZIP01*/*OsRE1*, Transcription factor HY5, putative, expressed	Post-embryonic development, signal transduction, secondary metabolism	Sequence-specific DNA binding transcription factor activity	Nucleus	-; [47]
4	*Os01g07910*/*BGIOSGA002284*	NADH-cytochrome b5 reductase, putative	Response to stress, response to abiotic stimulus	Binding, catalytic activity	Cell wall, mitochondrion	100% similar
5	*Os01g07920*	Prolyl 4-hydroxylase, putative	Protein modification process	Binding, catalytic activity	Golgi apparatus, vacuole, membrane	-; -
6	*Os01g07930*/*BGIOSGA002287*	Zinc finger C-x8-C-x5-C-x3-H (CCCH)-domain containing protein family, transcription factor	Biosynthetic process	Sequence-specific DNA binding transcription factor activity	-	100% similar
7	*Os01g07940*/*BGIOSGA002282*	AGC_PVPK_like_kin82y.3—ACG kinases include homologs to PKA, PKG and PKC	Reproduction, post-embryonic development, embryo development, protein modification process	Nucleotide binding, kinase activity	-	Deletion in *japonica* (126–131 bp); indica (924–932, 1201–3 bp); and SNPs
8	*Os01g07950*	OsGrx_S15.2—glutaredoxin subgroup II	Cellular homeostasis	Binding	Mitochondrion	-; -
9	*Os01g07960*	Acyl-protein thioesterase, Similar to Biostress-resistance-related protein	-	Hydrolase activity	-	-
10	*Os01g07980*	Ankyrin, putative, expressed. SGT1, suppressor of G2 allele of SKP1; Provisional	-	Binding	-	-; [48]
11	*Os01g08000*	Fibronectin type 3 and ankyrin repeat domains 1 protein	-	Protein binding	-	-; -
12	*Os01g08020*/*BGIOSGA002278*	Boron transporter protein, Bicarbonate transporter, eukaryotic domain containing protein.	Anion transport,	Borate efflux transmembrane transporter activity; inorganic anion exchanger activity	Integral component of membrane	Deletion in *japonica* (1–174 bp)
	*qGW1-1^BFSG^*	Chr1:5623500..5684500				
13	*Os01g10550*	*DEFL35*—Defensin-like DEFL family				-
14	*Os01g10580*/*BGIOSGA002958*	B-box (BBx) zinc finger family protein, transcription factor	Post-embryonic development, cellular component organization, secondary metabolic process, response to abiotic stimulus	Sequence-specific DNA binding transcription factor activity	Nucleoplasm	Deletion in *indica* (1–184; 197; 241; 841–846)
15	*Os01g10590*/*BGIOSGA002959*	*OsFTL8* FT-Like8 homologous to Flowering Locus T gene	Flower development, reproduction, post-embryonic development, response to abiotic stimulus	Protein binding, lipid binding	Nucleus, cytoplasm	Deletion in *japonica* (161–194 bp)
16	*Os01g10600*	*OsNIP1*;2 encoding Aquaporin protein, putative, expressed	Transport	Transporter activity	Membrane, plasma membrane	-; [49]
17	*Os01g10610*/*BGIOSGA002172*	*BRI1-EMS-SUPPRESSOR1*/ BRASSINOZANOL RESISTANT 1 (BES1/BZR1); transcriptional repressor family protein.	Brassinosteroids signaling-	-	-	Deletion in *indica* (1–93 bp) and SNPs (831:G/T, 836: T/C, 879: C/T); [50,51]
	*qGT1-1^Blink^*	Chr1:17993000..18054000				
18	*Os01g32780*/*BGIOSGA001545*	Universal stress protein domain-containing protein, UspA domain containing protein	Response to stress, response to molecule of fungal origin	-	-	100% similar; [52]
19	*Os01g32800*/*BGIOSGA001543*	Proteasome subunit, putative, expressed. PCI domain, also known as PINT motif (Proteasome, Int-6, Nip-1, and TRIP-15).	Protein metabolic process	Protein binding	Nucleus, intracellular, cytosol, proteasome complex	Deletion in *indica* (972–1004 bp)
20	*Os01g32870*	Heat shock protein DnaJ, Similar to Chaperone protein dnaJ 15 (Protein ALTERED RESPONSE TO GRAVITY) (AtARG1) (AtJ15) (AtDjB15).	Protein metabolic process, response to abiotic stimulus, protein binding tropism	-	-	-; -
21	*Os01g32880*	AP-3 complex subunit delta, Armadillo-type fold domain containing protein	Intra-Golgi vesicle-mediated transport, protein storage vacuole organization	Transporter activity, protein binding	Membrane, cytoplasm, Golgi apparatus	-; -
22	*Os01g32920*/*BGIOSGA003627*	ZOS1-08—C2H2 zinc finger protein, expressed, Transcription factor	Biosynthetic process	Sequence-specific DNA binding transcription factor activity	Intracellular	SNP689: T/C
23	*Os01g32930*/*BGIOSGA003628*	SGT1-specific (SGS) domain-containing protein	Embryo development, reproduction, post-embryonic development, protein binding, signal transduction, protein metabolic process, response to biotic stimulus	-	Cytosol	Deletion in *indica* (1–12, 270, 275–294 bp), *japoica* (479–484), SNPs (272: T/C, 319: C/A, 447: C/T, 504–505: GG/AC)
	*qGT2-1^Farm^*	Chr2:2088000..2151000				
24	*Os02g04630*	Sodium/calcium exchanger protein, putative, expressed. The Ca2+:Cation Antiporter (CaCA) Family (TC 2.A.19) proteins	Transport	Transporter activity	Cell, vacuole, membrane	-; -
25	*Os02g04640*/*BGIOSGA007162*	PHOSPHATE STARVATION RESPONSE 3 (OsPHR3), Myb-like DNA-binding domain containing protein, transcription factor	Nucleobase, nucleoside, nucleotide and nucleic acid metabolic process	Sequence-specific DNA binding transcription factor activity	-	100% similar; [53]
26	*Os02g04650*/*BGIOSGA007161*	Activator of 90 kDa heat shock protein ATPase homolog	Catabolic process	Enzyme regulator activity, protein binding	-	100% similar; -
27	*Os02g04660*	Arginine N-methyltransferase 5	Response to abiotic stimulus, protein modification process	Transferase activity	Cytosol	-; -
28	*Os02g04670*/*BGIOSGA007498*	Glucan endo-1,3-beta-glucosidase precursor	Carbohydrate metabolic process	Binding, hydrolase activity	Plasma membrane, membrane	Deletion in *japonica* (44–52 bp)
29	*Os02g04680*/*BGIOSGA007499*	Squamosa promoter-binding-like protein 3 (OsSPL3)—SBP-box gene family member, Transcription factor	Flower development, multicellular organismal development	Sequence-specific DNA binding transcription factor activity	Nucleus	100% similar; [54]
30	*Os02g04690*	Cycloartenol synthase	Multicellular organismal development, cellular component organization, lipid metabolic process	Catalytic activity	Vacuole	-; -
31	*Os02g04700*	tRNA synthetases class II domain-containing protein	Translation	Catalytic activity, nucleic acid binding	Cytosol, cytoplasm	-; -
32	*Os02g04710*	Cycloartenol synthase	Multicellular organismal development, cellular component organization, lipid metabolic process	Catalytic activity	Vacuole	-; -
33	*Os02g04725*	Dolichol phosphate-mannose biosynthesis regulatory protein	Macromolecule biosynthetic process	-	Cell, integral component of endoplasmic reticulum membrane	-; -
	*qLWR2^BFSG^*	Chr2:9970000..10030000				
34	*Os02g17350*/*BGIOSGA007951*	VPS-27, Hrs, and STAM (VHS) and GGA and Tom1 (GAT) domain-containing protein	Transport	Transporter activity	Golgi apparatus, plasma membrane	100% similar
35	*Os02g17360*/*BGIOSGA006711*	Restorer of fertility gene, Rf, pentatricopeptide repeat (PPR) repeat domain-containing protein	Mitochondrial cytoplasmic male sterility (CMS)	Nuclease activity	Plastid, mitochondrion	deletion in *indica* (1–84 bp); [55]
36	*Os02g17380*	Fertility restorer 2 (Rf2), Mitochondrial glycine-rich protein, Fertility restoration in LD-CMS	-	-	-	-; [56]
37	*Os02g17390*/*BGIOSGA007953*	ABNORMAL INFLORENSCENCE MERISTEM 1(MFP/AIM1); 3-hydroxyacyl-CoA dehydrogenase	Flower development, multicellular organismal development, post-embryonic development, lipid metabolic process	Catalytic activity	Plastid, cell wall, peroxisome	100% similar; [45]
38	*Os02g17400*/*BGIOSGA006709*	Leucine-rich repeat protein	Signal transduction, response to biotic stimulus, response to stress	-	Cell wall	Deletion in *indica* (96–101 bp)
39	*Os02g17460*	Tesmin/TSO1-like CXC domain-containing protein; transcrption factor	Biosynthetic process	Sequence-specific DNA binding transcription factor activity	-	-; -
	*qTGW6^BFSG^*	Chr6:28484608..28484625				
40	*Os06g46910*/*BGIOSGA023481*	ZOS6-07—C2H2 zinc finger transcription factor, expressed	Biosynthetic process	Sequence-specific DNA binding transcription factor activity	Intracellular	(SNP329: A/C; SNP445: T/C; SNP676: A/G; SNP1318: G/A); [57]
41	*Os06g46920*	Dihydroflavonol-4-reductase, NAD(P)-binding domain containing protein	Fatty acid catabolic process, gibberellin (GA) biosynthesis process, Seed dormancy process, GA-mediated signaling pathway	Cinnamyl-alcohol dehydrogenase activity, coenzyme binding, nucleotide binding, catalytic activity	-	-; -
42	*Os06g46930*/*BGIOSGA020659*	50S ribosomal protein L24, chloroplast precursor (CL24)	Pastid translation	Structural constituent of ribosome	Ribosome, plastid, large ribosomal subunit, chloroplast stroma	(SNP51: T/G; SNP282: A/G)
43	*Os06g46940*	*Os6bglu25*—beta-glucosidase homologue, similar to *Os3bglu6*, expressed	Carbohydrate metabolic process	Hydrolase activity, binding	Cell wall	-; [58]
44	*Os06g46950*/*BGIOSGA023482*	Carotenoid cleavage dioxygenase 1(OsCCD1), EF-hand calcium (Ca^2+^)-binding protein familyexpressed	Anatomical structure morphogenesis, cellular component organization, cell differentiation, multicellular organismal development	Calcium ion binding	-	100% similar; [59,60]
45	*Os06g46995*	Armadillo/beta-catenin repeat family protein, putative, expressed	-	Protein binding	-	-; -
46	*Os06g47000*/*BGIOSGA020655*	External NADH-Ubiquinone oxidoreductase 1, mitochondrial precursor, putative, expressed	Metabolic process	Catalytic activity	Membrane, mitochondrion	100% similar; -

## 3. Discussion

### 3.1. Grain Length, Width, and Thickness Are Closely Related to Thousand-Grain Weight but Not Length-to-Width Ratio

Understanding the correlation between factors helps quantify the strength of the direct relationship between them and figure out their affiliation [61]. Thousand-grain weight (TGW) is a determinant component of rice yield and is influenced by several factors, including grain length (GL), width (GW), and thickness (GT) [62,63,64]. In addition to grain-filling [65], it has been established that grain weight is determined by factors such as GL and GW, which contribute to enhancing the yield of rice [66,67]. Hence, the observed strong positive correlation between GW and TGW (R^2^ = 0.392 ***), and that between GL (R^2^ = 0.153 ***) or GT (R^2^ = 0.205 ***) and TGW would partially explain the shift in the TGW of the RIL population as shown in Figure 2M,N.

### 3.2. Genomic Estimated Breeding Value of RILs Population and Traits Heritability

The use of genomic selection (GS) in plant breeding has proven essential to increase the genetic gain of complex traits per unit of time and cost by enhancing the genomic estimated breeding value (GEBV) accuracies, through employing dense markers, and traits heritability [68]. GS also estimates the genetic merit of individuals (in this case the RILs) based on a large set of dense markers (here SNP Chip DNA makers) across the whole genome. GS then derives the GEBVs of all individuals in the breeding population based on their genotype and phenotype profiles and predicts those that are suitable for downstream breeding programs, relying on their actual performance [69]. Here, data obtained from GS analysis revealed the GEBV profile of RILs for thousand-grain weight, which is useful for downstream breeding using the best-performing RILs and associated SNP Chip DNA markers. It was interesting to see that GL (h^2^ = 0.915) and GW (h^2^ = 0.885), which were earlier shown to be closely related to TGW (h^2^ = 0.852), recorded high heritability scores. A study by Chen et al. [70] observed a high heritability for grain shape and weight, but environmental factors, including temperature, largely influence the phenotypic values of these traits.

### 3.3. The qGL1-1^BFSG^ QTL Harbors Genes Involved in Post-Embryonic Development and Reproduction

Grain length-related QTLs have been reported in Chr1 (*qGL1*), Chr2 (*qGL2.1, qGL2.2*), Chr3 (*GS3*), Chr4 (*qGL4*), Chr6 (*qGL6*), Chr7 (*qGS7, qGL7*), Chr8 (*qGL8.1*), Chr10 (*qGL10*), and Chr11 [70,71]. Here, we noted that the major QTL *qGL1-1^BFSG^* associated with the control of grain length in rice harbors genes proposed to be involved in reproduction, post-embryonic development, and embryo-development or protein-modification events (*Os01g07880* (HY5: elongated hypocotyl 5) and *Os01g07940* (AGC-PVPK)). The HY5 encodes a bZIP (basic leucine zipper) transcription factor highly conserved across plant species, and it is described as a central regulator of light signaling, acting as a pivotal regulator of light-dependent development [72]. The HY5 also functions in the regulation of nutrient uptake and utilization by controlling the expression of a large set of genes involved in nitrogen uptake and transport [73,74,75]. Other reports suggest the role of HY5 in light-mediated root growth [76], and sucrose efflux events (by inducing the expression of *SWEET11* and *SWEET12* (*SUCROSE TRANSPORTER*) [75,77]. In the same way, HY5 physically interacts with a group of B-box proteins (BBXs) [78,79,80,81] and other proteins [82] to regulate the expression of several target genes as well as multiple molecular and biological events.

In the same region (*qGL1-1^BFSG^*), a gene encoding a Zinc finger (CCCH) encoding a TF and two others encoding Ankyrin repeat domain-containing protein. Genes encoding the CCCH Zinc-finger protein have been proposed to regulate the adaptation of plants to abiotic stress [83,84,85]. Likewise, Ankyrin repeat domain-containing protein-encoding genes are thought to exclusively function to meditate protein–protein interactions and disease response [86].

### 3.4. The Grain Width, Thickness, and LWR-Associated QTLs qGW1-1^BSFG^, qGT1^Blink^, qGT2-1^Farm^ and qLWR2^BSFG^ Carry Genes Involved in Flower Development, Post-Embryonic Development and Reproduction

Several loci controlling GT, GW, and LWR have been reported under various growth conditions, and mapped to almost all chromosomes of rice (Chr1, 2, 3, 6–9, 11, 12) [2,70,71]. In the *qGW1^BF^* region, we noticed the presence of a gene encoding the *flowering time-like 8* locus (*OsFTL8*, *Os01g10590*), associated with flower development and reproduction. A previous report proposed that a member of the FTL family, *OsFTL4* (*Os09g33850*), regulates flowering time in rice in response to changing environmental conditions [87]. Likewise, a set of genes encoding a B-box (BBX) zinc finger protein (*Os01g40580*) or *OsNIP1* (*Os01g10600*, Aquaporin) are located within the *qGW1-1^BFSG^* region. Members of the BBXs family are a class of zinc finger proteins that encode transcription factors, and are mapped across the rice genome [88,89,90]. Among them, the *OsBBX14* (*Os05g11510*) was proposed to promote photomorphogenesis in rice [88]. In the same way, aquaporin is mainly associated with water movement in- and outside the cell. A study conducted by He et al. [91] revealed that *OsPIP1* encoding aquaporin interacts with other proteins to promote water uptake and seed germination. Furthermore, BES1/BZR1, a family of Brassinosteroids transcriptional regulators, were recently proposed to regulate plant development [92], kernel size in rice [50] and maize [51] through interaction with several proteins [93].

It was also interesting to see that genes located within the *qGT1^Blink^* locus or *qGT2-1^Farm^*, based on their predicted annotated functions, are associated with growth-related biological processes, including embryo development, reproduction, flower development (*OsSGT1*, *Os01g322890*; *OsSPL3*, *Os02g04660*), or transport, as well as transcriptional regulation (*ZOS1-08*, *Os01g32920*; PHR3, *Os02g04640*) [94].

As for the *qLWR2^BFSG^*, this QTL harbors genes with interesting annotated functions, including two genes (*Os02g17350* and *Os02g17380*, *OsRf2*) described as being involved in the restoration of fertility (cytoplasmic male sterility, CMS). The *Rf2* gene was earlier suggested to be involved in the mechanism for the restoration of fertility in CMS lines in rice [56].

### 3.5. The Grain Weight-Related QTL qTGW6^BFSG^ Harbors Genes Associated with Anatomical Structure Morphogenesis, Cell Differentiation, and Carbohydrate Metabolism

Thousand-grain weight (TGW) is controlled by several genetic loci. To date, many quantitative trait loci (QTLs) proposed to control TGW in rice have been identified, and mapped on all 12 chromosomes of rice, and a few genes have been functionally characterized. Multiple genetic and molecular aspects of plants affect grain weight, leading to dynamic changes in cell division, expansion, and differentiation [95].

The marker AX-115737727 is linked to the major QTL for TGW (*qTGW6^BFSG^*, Chr6: 28,484,619 bp) that coincides with the *qGW6-1^Blink^* QTL identified by the present study. We could mention the *Os06g46950* encoding the carotenoid cleavage dioxygenase 1 (CCD1) protein, the ZOS6-07 C2H2 Zinc finger TF (*Os06g46920*) or the *Os6bglu25* (*Os06g46940*, encoding the β-glucosidase homologue). A study by Ren et al. [58] suggested that a member of the β-glucosidase protein family, *Os06gGlu24* plays a role in seed germination and root elongation, while interacting with indole-3-acetic acid (IAA) and abscisic acid (ABA) signaling. Likewise, Ilg et al. [59] proposed the *CCD1* gene as being involved in the control of endosperm color in rice.

Although grain length, width, thickness, or thousand-grain weight are known to be controlled by multiple loci, genes harbored by *qGL1-1^BFSG^*, *qGT2-1^Farm^*, *qGW1-1^BFSG^*, or *qTGW6^BFSG^* share commonalities such as being involved in multicellular organismal development, flower development or reproduction, cell division or differentiation, among other annotations. It has been evidenced that TGW largely depends on GL, GW, and GT [96], in addition to grain filling ratio.

## 4. Materials and Methods

### 4.1. Plant Materials, Growth Conditions, and Phenotypic Measurements

A hundred and forty-three recombinant inbred lines (RILs), derived from a cross between Ilpum (*Oryza sativa* L. ssp. *japonica*) and Tung Tin Wan Hein1 (TTWH1, *Oryza sativa* L. ssp. *indica*) were used to conduct the experiments. Initially, pre-germinated seeds of RILs were sown and grown in 50-well trays until transplanting time. Then, healthy and vigorous four-week-old seedlings were transplanted (cropping season May to October 2022) in the experimental field (altitude: 11 m, 35°29′31.4″ N, and 128°44′30.0″ E), located at the National Institute of Crop Science (NICS), Department of Southern Area Crop Science, Paddy Crop Division, Rural Development Administration, Miryang, Republic of Korea. In the field, a total of 100 seedlings per RIL and parental lines were transplanted in four rows, with 25 plants per row and the spacing between and within the lines of 30 cm × 15 cm, respectively. The panicles for phenotypic observations of parental lines and RILs were harvested from the inside rows, excluding the border rows to avoid the border effects on the traits studied, and the competition between lines.

Soon after harvesting and postharvest processing, the grain size and shape-related phenotypes, including grain length (GL), grain width (GW), grain thickness (GT), grain length-to-width ratio (LWR, calculated as the GL divided by GW), and thousand-grain weight (TGW) were measured. The GL, GT, GW, and LWR were measured or calculated using the SmartGrain v.1.2 (copyright© 2010–2012, Takanari TANABATA, Tsukuba, Japan; http://phenotyping.image.coocan.jp, accessed on 12 April 2023). Before analysis, 100 rice seeds, with a label that helps identify the RIL under analysis, were placed on the Canon scanner 5600F model using a typical rectangular rice seed dispenser (Appendix A), and the dispenser was removed thereafter. Seeds were scanned and the image saved in an appropriate folder, for further processing (Appendix A). Prior to analyzing the phenotype of grains, basic settings were performed, such as the selection of seed detection sensitivity strength, picking seed and background colors by right-clicking inside the imported image, determining the scale bar, etc. To analyze, we clicked on “Analyze” in the title bar, selected “Analyze area” in the drop window, and selected the target region on the open image to analyze. Final quality control was performed to ensure the accuracy of the measurement as follows: Set [Disable/Enable] (right click on the mouse) to unselect or select seeds on the image, followed by exporting as Excel “csv.” Format (Appendix A).

The GT was measured manually using a digital Vernier Caliper (CD-20CP, Mitutoyo Corp, Tokyo, Japan). However, the TGW was calculated as the [(average grain weight of 100 dehulled seeds/the number of samples (n)) × 10].

### 4.2. Frequency Distribution, Correlation Analysis, Quantile–Quantile Plots, Kinship Matrix

To assess the frequency distribution of traits, generate the box plots, and investigate the Pearson correlation between the target traits, GraphPad Prism 7.0 (© 1992–2016 GraphPad Software, Inc., Odesa, Ukraine) was used. The pairwise kinship matrix, also known as the co-ancestry or half relatedness, the principal component analysis (PCA) plot, and the Quantile–Quantile (Q–Q) plots were generated from the GAPIT package using R software. The SNP density plot was generated using filtered SNP Chip DNA markers with their relative *p*-values (GWAS results in .csv file) using the below script:

install.packages(‘CMplot’)

library(CMplot)

head(my_data)

CMplot(my_data,type=“p”,plot.type=“d”,bin.size=1e6,chr.den.col=c(“darkgreen”, “yellow”, “red”),file=“jpg”,file.name=““,dpi=300, main=“SNP Chip Markers Density”,file.output=TRUE,verbose=TRUE,width=9,height=6)

Genomic Selection or Prediction Analysis

To investigate the genetic merit of the RILs for specific target traits, a genomic prediction or selection analysis was conducted as described by Zhang et al. [97]. The genomic best linear unbiased prediction (gBLUP), commonly used for the genomic selection based on mixed model (MLM), and having a higher prediction accuracy for traits controlled by a large number of genes was used perform the genomic selection Yu et al. [98]. The genotype data were converted from the Haplotype Map (HapMap) format to numerical (see R script below) prior to performing the analysis.

To convert HapMap to numerical format:

myG <-fread(“file:///D:genotype data location.txt”, head = FALSE)

myGAPIT <- GAPIT(G=myG, output.numerical=TRUE)

myGD= myGAPIT$GD

myGM= myGAPIT$GM

To conduct a genomic prediction:

myY<-read.csv(“phenotype file location pathway.csv”, sep = “,”)

myGD=read.csv(“numerical genotype file location pathway.csv”, sep = “,”)

myGM=read.csv(“markers file location pathway.csv”, sep = “,”)

set.seed(99163)

GAPIT.Validation(

Y=myY[,1:2],

model=c(“gBLUP”),

GD=myGD,

GM=myGM,

PCA.total=3,

file.output=T,

nfold=5

The GS/GP of the inference groups (based on the ties with corresponding groups in the reference panel) was derived from Henderson’s formula as follows:*u*_I_ = K_IR_K_RR_^−1^*u*_R_,
where K_RR_ is the variance–covariance matrix for all groups in the reference panel, K_RI_ is the covariance matrix between the groups in the reference and inference panels, K_IR_ is the covariance matrix between the groups inference and reference panels, and *u*_R_ is the predicted genomic values of the individuals in the inference group. To assess the reliability of the genomic prediction, the following formula is used:Reliability = 1 − PEV/σ^2^_a_,
where PEV is the prediction error variance, representing the diagonal element in the inverse left-hand side of the mixed model equation, and σ^2^_a_ is the genetic variance.

### 4.3. Genome-Wide Association Study (GWAS) Analysis

To assess the association between potential genetic loci and the traits of interest at the whole genome level, we performed a Genome-Association Study (GWAS) employing the Genome Association and Prediction Integrated Tool (GAPIT) version 3 [99] with multiple models with enhanced power and accuracy for genome association. The GAPIT models used in this study include the Bayesian-information and Linkage-disequilibrium Iteratively Nested Keyway (BLINK) [100], the Fixed and random model Circulating Probability Uniform (FarmCPU) [101], Settlement of MLM Under Progressively Exclusive Relationship (SUPER), and the General Linear Model (GLM) [102]. FarmCPU and SUPER supports genomic selection, while BLINK and GLM are commonly used for breeding through marker-assisted selection (MAS).

To perform a GWAS analysis, the below R script was used, after setting the results directory (*setwd()*), installing (install.packages (“package name”) and launching all necessary packages and their libraries (package name)), installing the GAPIT source code, importing the genotype (*geno.raw <- fread(“file:///D:/... .csv or .txt*) and phenotype (*myY <- fread(“file:///D:/... .csv or .txt*) files, and performing initial data quality control:

my_GAPIT <- GAPIT(Y=myY, G=myG, model=c(“SUPER”, “FarmCPU”, “BLINK”), PCA.total=3, SNP.MAF = 0.05, Multiple_analysis=TRUE)

### 4.4. In Silico Analysis for Gene Ontology Search

GWAS results provided useful information on novel genetic loci for grain size and shape in rice. The physical positions of associated significant SNP Chip markers were utilized to uncover the identity of genes harbored by the target genetic loci for more insights. To achieve that, we conducted a search using the browser of the Rice Genome Annotation Project database (http://rice.uga.edu/cgi-bin/gbrowse/rice/#search, accessed on 7 September 2023) and PlantPAN 3.0 (http://plantpan.itps.ncku.edu.tw/plantpan3/search.php?#results, accessed on 7 September 2023) for each specific gene locus ID. Genes encoding similar domain-containing proteins were searched in the literature (https://funricegenes.github.io/geneKeyword.table.txt, accessed on 7 September 2023).

To assess the degree of sequence similarity of genes found in major QTLs for grain traits, the coding sequence (CDS) of each gene in the *japonica* group was aligned with that of its orthologous gene in the *indica* group. The respective CDS of target gene locus IDs (i.e., LOC_Os01g10580: Nipponbare database (http://rice.uga.edu/analyses_search_locus.shtml, accessed on 11 September 2023), and BGIOSGA002958: *indica* database (https://plants.ensembl.org/Oryza_indica/Info/Index, accessed on 11 September 2023) were obtained, and aligned using the ClustalW multiple alignment feature in Bioedit sequence Alignment Editor Software (Copyright © 1997–2013 Tom Hall, USA) [103].

## 5. Conclusions

Rice grain-related traits are controlled by multiple genetic loci in plants. Grain length, width, and thickness determine the thousand-grain weight, thus influencing rice yield. A total of 43 QTLs associated with grain size, shape, or weight in rice, are distributed across almost all rice chromosomes. GWAS results show seven SNP Chip makers (co-detected by both BLINK and FarmCPU) with strong association with grain length on Chr1–3, 5, 7, and 11, with *qGL1-1^BFSG^* explaining 65.2–72.5% of the observed phenotypic variance for grain length. In addition, one (*qGW1-1^BFSG^*) out of fourteen QTLs for grain width was co-detected by all four GAPIT models on Chr1. The *qGW1-1^BFSG^* explains 15.5–18.9% of PVE. Likewise, either BLINK or FarmCPU identified three QTLs for grain thickness. Two of them explain 74.9% (*qGT1^Blink^*) and 54.9% (*qGT2-1^Farm^*) of the observed PVE. Regarding length-to-width ratio, the *qLWR2^BFSG^*, detected by all GAPIT models, explains about 15.2–32%) for LWR. As for thousand-grain weight, the *qTGW6^BFSG^* QTL coincided with *qGW6-1^Blink^* for grain width and explained 32.8–54% of PVE. Putative candidate genes pooled from co-detected regions by all four GAPIT models have interesting annotated functions, and either associated with flower development, reproduction, post-embryonic development, carbohydrate metabolisms, or transcription regulation. Downstream functional studies, through the use of genetic engineering approaches or mutagenesis, would help elucidate the molecular functions of the candidate genes. The major QTLs for each grain trait can serve for downstream marker-assisted selection based on genome selection results.

## Figures and Tables

**Figure 1 plants-12-04044-f001:**
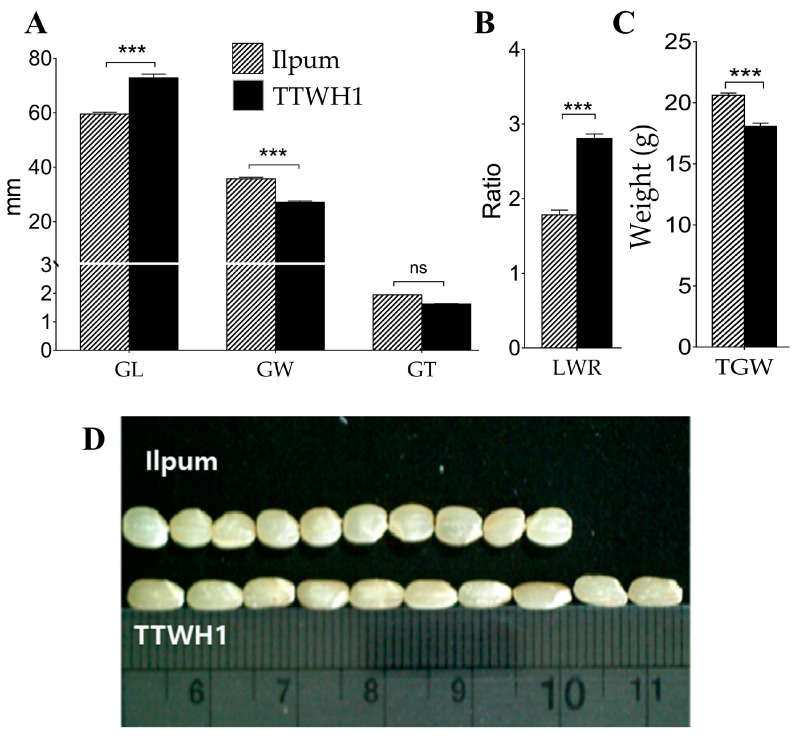
Differential phenotypic difference between parental lines. (**A**) Comparison of rice grain trait values (grain length, width, and grain thickness) of Ilpum (*japonica*) and Tung Tin Wan 1 (*indica*), (**B**) grain length-to-width ratio, (**C**) thousand-grain weight of parent lines, and (**D**) visualization of grain phenotypes of parental lines (Ilpum, upper side and TTWH1 downside). Bars (with hatches and those filled in black) are mean values (*n* = 10) ± SE. *** *p* < 0.001, ns non-significant.

**Figure 2 plants-12-04044-f002:**
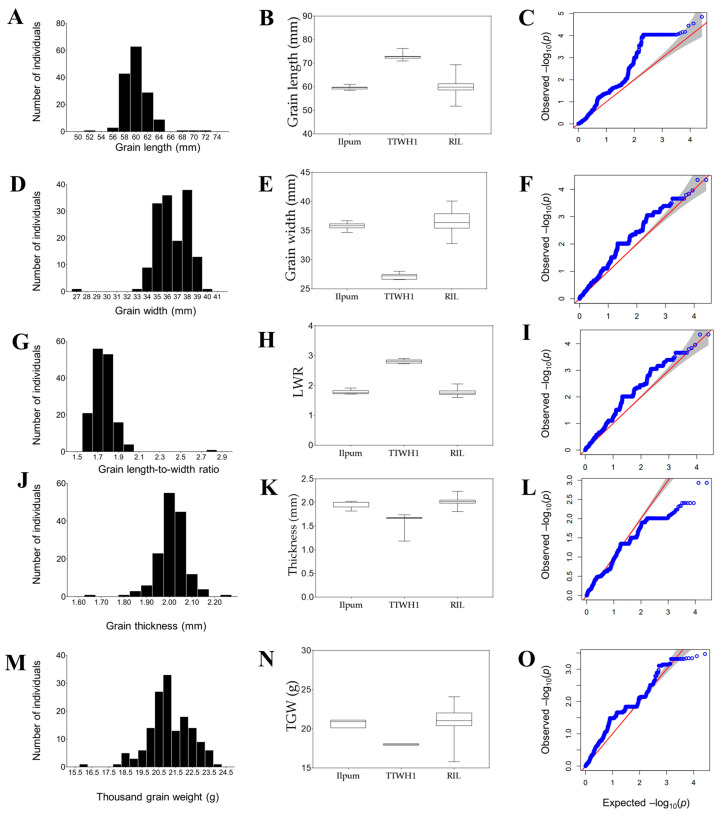
Frequency distribution of traits, box plots, and Quantile–Quantile (Q–Q) plot. (**A**) Frequency distribution for grain length, (**D**) grain width, (**G**) length-to-width ratio, (**J**) grain thickness, (**M**) thousand-grain weight. (**B**) box plots showing the shift in grain length of the recombinant inbred lines relative to their parental lines, (**E**) grain width, (**H**) grain length-to-width ratio, (**K**) grain thickness, (**N**) thousand-grain weight, (**C**) Quantile–Quantile (Q–Q) plot for grain length, (**F**) grain weight, (**I**) grain length-to-width ratio, (**L**) grain thickness, and (**O**) thousand-grain weight, with the y-axis representing the quantile in the sample (RIL population) and the x-axis denoting data falling in the probability distribution. The blue thick lines in the Q–Q plots represent the observed quantile (set of values of a variate that divides a frequency distribution into equal groups, each containing the same fraction of the total population) plotted against the expected quantile (red line).

**Figure 3 plants-12-04044-f003:**
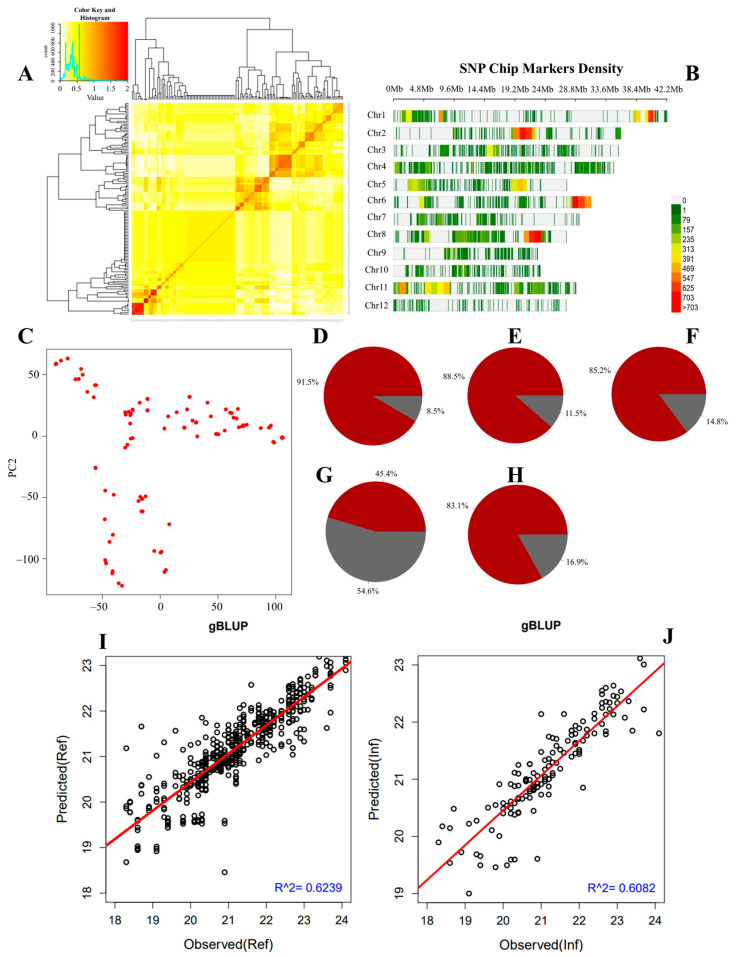
Kinship matrix, marker density, PCA, heritability, and genome selection results. (**A**) heat map showing the relatedness or the level of co-ancestry of the population, (**B**) density map of SNP Chip DNA markers, (**C**) principal component analysis (PCA), (**D**) narrow sense heritability of grain length, (**E**) grain width, (**F**) grain thickness, (**G**) grain length-to-width ratio, (**H**) thousand-grain weight, and (**I**,**J**) results of the genome selection analysis that predict the genomic estimated breeding value (GEBV) of individuals in the RIL population in the reference group (Ref) and the inference group (Inf).

**Figure 4 plants-12-04044-f004:**
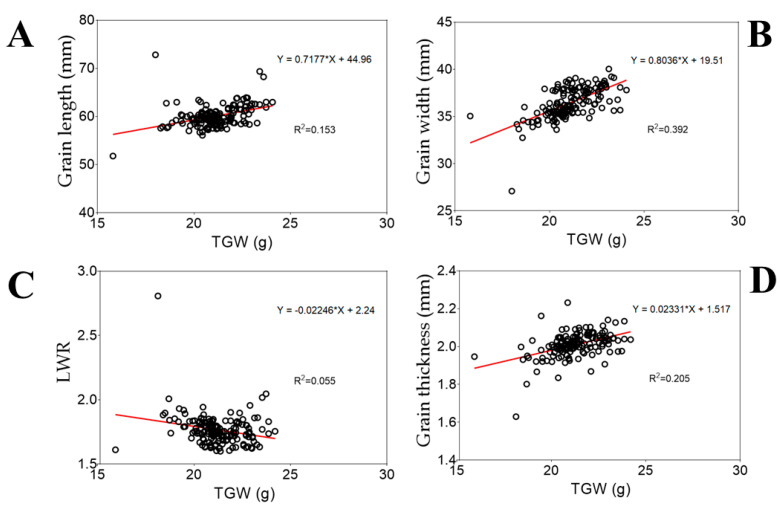
Pearson correlation analysis results between traits. (**A**) Correlation results between grain length and thousand-grain weight, (**B**) grain width and thousand-grain weight, (**C**) length-to-width and thousand-grain weight, and (**D**) grain thickness and thousand-grain weight. The closer the data points come to forming a straight line (around the red line), the higher the correlation between the two variables, or the stronger the relationship.

**Figure 5 plants-12-04044-f005:**
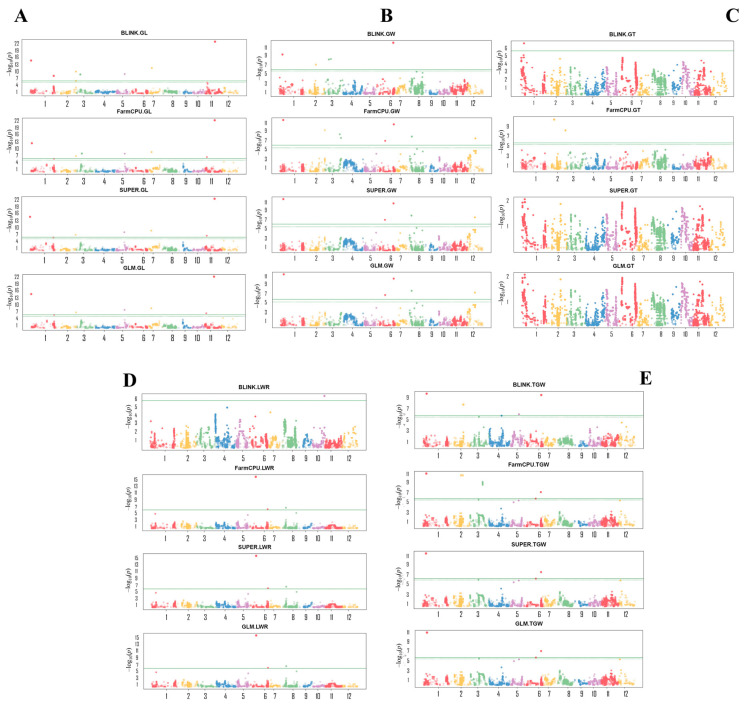
Detected significant SNP Chip markers by GWAS. Genome-wise Manhattan plots of showing significant SNP Chip DNA markers detected by BLINK, FarmCPU, SUPER, and/or GLM GAPIT models, with their associated with (**A**) grain length, (**B**), grain width, (**C**) grain thickness, (**D**) grain length-to-width ratio, and (**E**) thousand-grain weight.

**Figure 6 plants-12-04044-f006:**
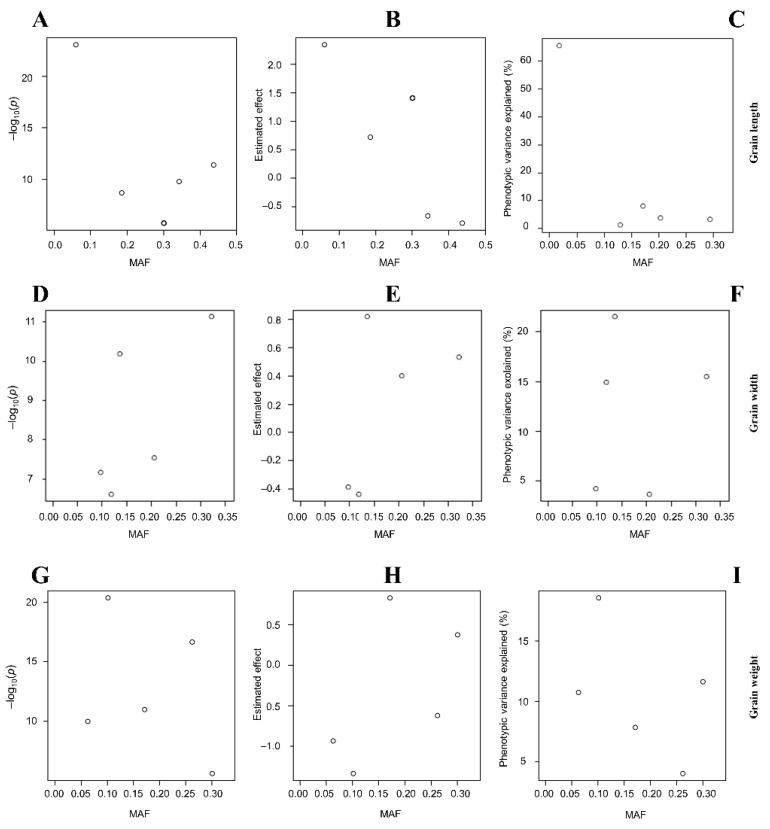
QTL logarithm of the odds, estimated effects and phenotypic variance explained. Logarithm of the odds (LOD) scores for significant SNP Chip DNA markers linked to (**A**) grain length, (**D**) width, and (**G**) thousand-grain weight. (**B**,**E**,**H**) Estimated effects of QTLs, and (**C**,**F**,**I**) phenotypic variance explained (PVE) values of QTLs for grain length, width, and thousand-grain weight, respectively. −log10(*p*) is the negative logarithm base 10 of the *p*-value denoting the logarithm of the odds (LOD) of SNP Chip markers linked to detected genetic loci for each trait. MAF plotted in the x-axis of all panels indicates the minor allelic frequency referring to the lower allelic frequency at a given genetic locus in the RIL population.

**Table 1 plants-12-04044-t001:** Detected grain traits-related QTLs by GAPIT model.

Traits/QTLs	SNP Markers	Chr	Position (bp)	PVE (%)	GWAS-GAPIT Models				Allele
**Grain Length**									
*qGL1-1^BFSG^*	AX-95918134	1	3,820,526	72.5	BLINK	FarmCPU	SUPER	GLM	TTWH1
*qGL11-1 ^BFSG^*	AX-274862201	11	16,356,105	31.9	BLINK	FarmCPU	SUPER	GLM	Ilpum
*qGL2-1^BFSG^*	AX-115751685	2	35,100,558	22.9	BLINK	FarmCPU	SUPER	GLM	TTWH1
*qGL3-1^BF^*	AX-154437636	3	2,253,428	5.5	BLINK	FarmCPU	-	-	Ilpum
*qGL2-2^BF^*	AX-154880023	2	35,133,528	3.1	BLINK	FarmCPU	-	-	TTWH1
*qGL7-1^BFSG^*	AX-153903748	7	6,690,089	2.6	BLINK	FarmCPU	SUPER	GLM	TTWH1
*qGL1-2^BF^*	AX-279584700	1	41,489,588	1.7	BLINK	FarmCPU	-	-	TTWH1
*qGL5^BFSG^*	AX-282746698	5	16,169,537	1.4	BLINK	FarmCPU	-	-	Ilpum
*qGL11-2^FSG^*	AX-115751092	11	2,823,622	14.0	-	FarmCPU	SUPER	GLM	Ilpum
*qGL3-2^Farm^*	AX-154551783	3	8,242,439	8.0	-	FarmCPU	-	-	TTWH1
**Grain Width**									
*qGW1-1^BFSG^*	AX-273945773	1	5,623,288	18.9	BLINK	FarmCPU	SUPER	GLM	Ilpum
*qGW2-1^Blink^*	AX-279699609	2	10,805,604	14.9	BLINK	-	-	-	TTWH1
*qGW1-2^BF^*	AX-115791785	1	43,103,625	8.8	BLINK	FarmCPU	-	-	TTWH1
*qGW1-3^Blink^*	AX-281116133	1	20,864,932	6.9	BLINK	-	-	-	Ilpum
*qGW6-1^BF^*	AX-115737727	6	28,484,619	6.2	BLINK	FarmCPU	-	-	Ilpum
*qGW3-1^Blink^*	AX-154073979	3	7,895,651	4.8	BLINK	-	-	-	Ilpum
*qGW3-2^Blink^*	AX-115811160	3	14,888,685	4.1	BLINK	-	-	-	Ilpum
*qGW6-2^FSG^*	AX-273990782	6	13,986,482	14.9	-	FarmCPU	SUPER	GLM	TTWH1
*qGW1-4^Farm^*	AX-280898927	1	2,483,022	9.4	-	FarmCPU	-	-	TTWH1
*qGW12^FSG^*	AX-284265976	12	13,013,702	4.2	-	FarmCPU	SUPER	GLM	TTWH1
*qGW8 ^FSG^*	AX-115796459	8	3,875,546	3.7	-	FarmCPU	SUPER	GLM	Ilpum
*qGW2-2^Farm^*	AX-279994820	2	35,461,009	3.5	BLINK	FarmCPU	-	-	TTWH1
*qGW2-3^Farm^*	AX-154042022	2	24,704,256	3.4	BLINK	FarmCPU	-	-	Ilpum
*qGW3-3^Farm^*	AX-154797543	3	2,973,374	1.2	BLINK	FarmCPU	-	-	Ilpum
**Grain thickness**									
*qGT1^Blink^*	AX-279261704	1	18,023,142	74.9	BLINK	-	-	-	TTWH1
*qGT2-1^Farm^*	AX-154787777	2	2,118,477	54.9	-	FarmCPU	-	-	TTWH1
*qGT2-2^Farm^*	AX-154913392	2	25,105,471	5.3	-	FarmCPU	-	-	Ilpum
**Length-to-Width Ratio**									
*qLWR10^Blink^*	AX-115835839	10	22,038,978	26.5	BLINK	-	-	-	Ilpum
*qLWR2^BFSG^*	AX-274833045	2	10,000,097	15.2	BLINK	FarmCPU	SUPER	GLM	TTWH1
*qLWR1-1^BF^*	AX-154960834	1	1,595,394	13.5	BLINK	FarmCPU	-	-	TTWH1
*qLWR1-2^Blink^*	AX-115737888	1	600,441	10.7	BLINK	-	-	-	TTWH1
*qLWR3^Blink^*	AX-154834762	3	8,098,398	6.9	BLINK	-	-	-	TTWH1
*qLWR6-1^FSG^*	AX-115851421	6	10,178,858	30.5	-	FarmCPU	SUPER	GLM	Ilpum
*qLWR6-2^FSG^*	AX-155522120	6	30,842,264	9.4	-	FarmCPU	SUPER	GLM	TTWH1
*qLWR8^FSG^*	AX-154176130	8	5,398,451	5.9	-	FarmCPU	SUPER	GLM	TTWH1
**Thousand-grain weight**									
*qTGW6^BFSG^*	AX-115737727	6	28,484,619	32.8	BLINK	FarmCPU	SUPER	GLM	Ilpum
*qTGW2-1^BF^*	AX-279699609	2	10,805,604	18. 6	BLINK	FarmCPU	-	-	TTWH1
*qTGW3-2^Farm^*	AX-123153600	3	7,887,961	13.9	-	FarmCPU	-	-	Ilpum
*qTGW1-1^Blink^*	AX-154298059	1	5,644,298	11.6	BLINK	-	-	-	Ilpum
*qTGW3-1^BF^*	AX-154471576	3	15,332,432	10.7	BLINK	FarmCPU	-	-	TTWH1
*qTGW2-2^BF^*	AX-154096541	2	10,773,042	7.8	BLINK	FarmCPU	-	-	Ilpum
*qTGW1-2^BFSG^*	AX-154333920	1	5,860,250	4.9	BLINK	FarmCPU	SUPER	GLM	Ilpum
*qTGW1-3^BF^*	AX-154810092	1	42,931,550	4.03	BLINK	FarmCPU	-	-	TTWH1

GL: grain length, GW: grain width, GT: grain thickness, LWR: length-to-width ratio, and TGW: thousand-grain weight. Chr: chromosome, MAF: minor allelic frequency, nobs: number of observations, PVE: phenotype variance explained. *qTrait^Blink^*: QTL detected by BLINK only, *qTrait^Farm^*: QTL detected by FarmCPU only, *qTrait^FSG^*: QTL co-detected by FarmCPU, SUPER, and GLM, *qTrait^BFSG^*: QTL co-detected by BLINK, FarmCPU, SUPER, and GLM.

## Data Availability

Data are contained within the article and Appendix A.

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
