# Peer review of "Rice (*Oryza sativa* L.) Grain Size, Shape, and Weight-Related QTLs Identified Using GWAS with Multiple GAPIT Models and High-Density SNP Chip DNA Markers"

_plants, 2023, doi:10.3390/plants12234044_

Round 1

Reviewer 1 Report

Comments and Suggestions for Authors

I have highlighted some words/phrases that need clarification or correction. The study uses 143 RILs and has appropriately addressed the questions. The locusID OSgxxxx (highlighted in yellow) is a mistake or meant to represent several loci compared? 

Comments on the Quality of English Language

mInor problems marked in text.

Author Response

Reviewer 1

I have highlighted some words/phrases that need clarification or correction. The study uses 143 RILs and has appropriately addressed the questions.

We are thankful to the reviewer for his valuable comments and the time he took to review our manuscript. We have tried to improve our manuscript based on the concerns raised by the reviewer, and almost all the comments have been addressed.

The locus ID OSgxxxx (highlighted in yellow) is a mistake or meant to represent several loci compared? 

The locus ID in line 222 is a example of the structure of gene ID in japonica and iIndica databases. We have provided a real example for more clarity.

Minor problems are marked in the text.

We have revised the manuscript and corrected typos and grammatical errors in the text as suggested.

Reviewer 2 Report

Comments and Suggestions for Authors

It's a good study, but just ask why you did not distinguish between RILs and their parental lines in all the data analysis.

Comments on the Quality of English Language

Minor editing for English is required 

Author Response

Reviewer 2

It's a good study, but just ask why you did not distinguish between RILs and their parental lines in all the data analysis.

We appreciate the concern raised by the reviewer. We are sorry for this impression. In lines 97-105, we described the phenotype of parents and their differences (Figure 1). Lines 106-113 focus on the description of RIL phenotypes relative to their parental lines. We tried to see the changes or shifts in the RILs phenotypes in line with parent 1 or parent 2 as well as their frequency distribution (Figure 2). As for other data such as GWAS, parental data are not commonly included in the analysis.

Minor editing for English is required 

We have revised the manuscript for English as suggested.

Reviewer 3 Report

Comments and Suggestions for Authors

One: The experimental design is unreasonable

(1) Insufficient treatment conditions for experimental design. Scientific experimental design should have at least 3 treatments and 3 repetitions; For example: 2 years 2 regions, a total of 4 processing, each processing 3 repetitions. However, this article only has 1 year 1 region data, and the number of repetitions is not specified.

(2) Unreasonable group selection. GWAS analysis generally selects natural populations based on unrelated individuals as the study population, and the selection of natural populations generally meets the following criteria: having rich genetic variation, having a wide range of phenotypic differences, having a broad genetic basis of varieties, and having geographical representation. However, the population selection in this paper is not only recombinant inbred lines, but also does not have a wide range of phenotypic differences.

Two: The result analysis is not reliable

(1)Line:105 “In addition, we observed a normal distribution for grain length”

Has the "normal distribution test" been carried out, and where are the results of the analysis?

(2) Where is the Quantile–Quantile (Q–Q) plot? I saw the Manhattan plots analyzed by GWAS, but I did not see the Q-Q plot, the key indicator data corresponding to the Manhattan plot.

(3) What is the interval range of QTL?

(4) The screening process of candidate genes is only the "functional analysis" of genes, which is too simple and unreliable.

Three: The introduction is not clear

(1) There are six paragraphs in the "Introduction", which are redundant and unclear.

2) Line:139-141. The description of the picture does not correspond to the actual picture.

Author Response

Reviewer 3

One: The experimental design is unreasonable

We appreciate the concern raised by the reviewer.

(1) Insufficient treatment conditions for experimental design. Scientific experimental design should have at least 3 treatments and 3 repetitions; For example: 2 years 2 regions, a total of 4 processing, each processing 3 repetitions. However, this article only has 1 year 1 region data, and the number of repetitions is not specified.

We would like to share that during the development of any population or lines, the stability of the population is confirmed generation after generation until fixed, stable, and homozygous lines are obtained. This population has been employed in other studies to investigate genetic loci for disease control such as Bakanae and bacterial leaf blight, with Kompetitive Allelic-PCR markers (KASP). Here, we employed a complex analysis using multiple GAPIT models for several grain traits.

The planting conditions in the field are now specified in the text (Lines 115-119). The measurement of grain phenotypes is done on a few panicles randomly selected in the field, not on border lines to avoid the border effect. Four rows of 25 plants each (100 plants) were grown under regular water and fertility management.

(2) Unreasonable group selection. GWAS analysis generally selects natural populations based on unrelated individuals as the study population, and the selection of natural populations generally meets the following criteria: having rich genetic variation, having a wide range of phenotypic differences, having a broad genetic basis of varieties, and having geographical representation. However, the population selection in this paper is not only recombinant inbred lines but also does not have a wide range of phenotypic differences.

We appreciate the concern raised by the reviewer. While acknowledging that the natural population has been widely used for GWAS, it is equally true that many other population types have been largely used to investigate genetic loci controlling several agronomic traits in plant species employing GWAS [1,2] or QTL mapping, and in some other cases a combined strategy GWAS–QTL mapping in various DNA marker systems [3-5].

Figure 3A is the kinship matrix denoting the relatedness of individuals based on genetic markers. This data suggests the genetic variability of the population used (RIL) and their low relatedness. This aspect is described in lines 118-121. RILs showed phenotypic variance for studied traits. Grain traits are not like other growth-related parameters such as plant height, culm length, etc. were huge differences expected in the field between breeding lines. Nevertheless, i.e. grain width and TGW indicate (Figures 2D,J) have a wide range of phenotypic divergence. In other traits as well, from the error bars, we could read the existence of lines with much higher or much lower phenotypes compared to their parental lines or within the RILs.

Two: The result analysis is not reliable

We appreciate the opinion of the reviewer.

(1)Line:105 “In addition, we observed a normal distribution for grain length”

Has the "normal distribution test" been carried out, and where are the results of the analysis?

(2) Where is the Quantile–Quantile (Q–Q) plot? I saw the Manhattan plots analyzed by GWAS, but I did not see the Q-Q plot, the key indicator data corresponding to the Manhattan plot.

When performing the frequency distribution of traits, as part of statistical analysis, basic parameters are determined. The GraphPad 7 Prism software used in this study to perform descriptive statistics was employed to assess the frequency distribution. The skewness and normal distribution of traits for the RILs are reliable.

The Q-Q plot is in the panel K of Figure 2. This is a combined Q-Q plot for all traits (see the legend)

(3) What is the interval range of QTL?

The genetic loci for the studied traits are given in the manuscript. GWAS results provide more insights and data are presented in detail in Tables with significant SNP chip DNA markers associated with the trait of interest. The results format for GWAS is different from that of linkage mapping software such as IcIMapping, etc. The physical positions of signifcant markers are used to identify genes within the region.

(4) The screening process of candidate genes is only the "functional analysis" of genes, which is too simple and unreliable.

We appreciate the concern raised by the reviewer. We would like to share that this study does not intend to functionally characterize any of the putative candidate genes identified. However, we must acknowledge that the advent of sequencing technologies and the use of bioinformatics provided useful information of genes and their predicted annotated functions. Here, we only utilized the available information accessible to the public using physical positions of detected genetic regions. We clearly stated that funcitonal studies would help elucidated the molecular functions of the candidate genes (243-245).

Three: The introduction is not clear

(1) There are six paragraphs in the "Introduction", which are redundant and unclear.

We have tried to revised the introduction to avoid redundancy and to improve the readability of the text.

2) Line:139-141. The description of the picture does not correspond to the actual picture.

We apologize for the inconvenience. We have corrected the caption of Figure 2 as suggested.

Reviewer 4 Report

Comments and Suggestions for Authors

This manuscript describes the study of grain morphology QTLs in rice. The work is presented well, thoroughly described and the research appears to be robust and scientifically sound. There is room for some improvement in the discussion and conclusion sections to connect more with the reader by presenting more of a narrative and conveying more sense of why this work is important. I am less sure than the authors that there is a value in employing genomic selection for highly heritable traits.

Comments on the Quality of English Language

The manuscript is readable, but there is room for improvement in the quality of English language. There are some areas where the communication is hampered by the writing quality and a few instances where an incorrect word is used, for example, in the introduction L 43-46: "Despite the increase in population growth estimated to.." Despite is I think the opposite of what you mean to say here. "Due to the increase in population..." 

Author Response

Reviewer 4

This manuscript describes the study of grain morphology QTLs in rice. The work is presented well, thoroughly described and the research appears to be robust and scientifically sound.

There is room for some improvement in the discussion and conclusion sections to connect more with the reader by presenting more of a narrative and conveying more sense of why this work is important. I am less sure than the authors that there is a value in employing genomic selection for highly heritable traits.

We thank the reviewer for his valuable comments and suggestions to improve our manuscript.

We would like to share that genomic selection for enhanced breeding efficiency for crop improvement is used in agronomically important traits with different heritability [6,7], including as a decision support tool [8]. It is suggested that traits with a high heritability has good prediction accuracies, while those with low heritability affects the prediction accuracy. Trait with low heritability would require a larger training population in order to attain the prediction accuracty as in the case of traits with moderate to high heritability [6].

The manuscript is readable, but there is room for improvement in the quality of English language. There are some areas where the communication is hampered by the writing quality and a few instances where an incorrect word is used, for example, in the introduction L 43-46: "Despite the increase in population growth estimated to.." Despite is I think the opposite of what you mean to say here. "Due to the increase in population..." 

We appreciate the comments from the reviewer. We have revised the manuscript and tried to improve the quality of English where necessary.

Academic Editor

Authors investigated the novel QTLs for grain shape and weight by combination of GWAS and high density SNP Chip Markers. Three referees gave positive score among four referees, this manuscript can be acceptable to publish in this journal. Before acceptance, please revise the manuscript carefully according to the comments raised by all referees. Also, please amend the Figure presentation nicely to attract readers.

We appreciate the valuable comments and suggestions made by the Academic Editor to improve the quality of Figure and some areas of the manuscript.

The balance of the size of Figures are not well, please arrange them more balanced way.

We have tried to improve the balance of Figures in the manuscript as suggested.

1) Use more bigger font to readable size in all Figures, especially, Figure2K, Figure 3, Figure 5.

We have increased the font of values in figures for a better readability.

2) Enlarge the size of Figures to see the details in Figure 3A, and the Manhattan plots in Figure 5A-E.

We have tried to improve the quality of Figures as suggested.

  1. Budhlakoti, N.; Kushwaha, A.K.; Rai, A.; Chaturvedi, K.; Kumar, A.; Pradhan, A.K.; Kumar, U.; Kumar, R.R.; Juliana, P.; Mishra, D.J.F.i.G. Genomic selection: a tool for accelerating the efficiency of molecular breeding for development of climate-resilient crops. 2022, 13, 832153.
  2. Roy, A.; Purkaystha, S.; Bhattacharyya, S.J.H.E.; Molecular, P.R.; Aspects, F. Advancement in Molecular and Fast Breeding Programs for Climate-Resilient Agriculture Practices. 2021, 73-98.
  3. Prasanna, B.; Cairns, J.; Xu, Y. Genomic tools and strategies for breeding climate resilient cereals. In Genomics and Breeding for Climate-Resilient Crops: Vol. 1 Concepts and Strategies; Springer: 2013; pp. 213-239.